# Prognostic Role of Prolactin-Induced Protein (PIP) in Breast Cancer

**DOI:** 10.3390/cells12182252

**Published:** 2023-09-11

**Authors:** Natalia Sauer, Igor Matkowski, Grażyna Bodalska, Marek Murawski, Piotr Dzięgiel, Jacek Calik

**Affiliations:** 1Faculty of Pharmacy, Wroclaw Medical University, 50-556 Wroclaw, Poland; 2Old Town Clinic, 50-127 Wroclaw, Poland; 3Jan Mikulicz-Radecki University Teaching Hospital, Borowska 213, 50-556 Wroclaw, Poland; igor.matkowski2@gmail.com; 4Faculty of Medicine, Wroclaw Medical University, 50-556 Wroclaw, Poland; grazyna.bodalska@student.umw.edu.pl; 51st Department and Clinic of Gynecology and Obstetrics, Wroclaw Medical University, 50-556 Wroclaw, Poland; marek.murawski@umw.edu.pl; 6Division of Histology and Embryology, Department of Human Morphology and Embryology, Wroclaw Medical University, T. Chalubinskiego 6a, 50-368 Wroclaw, Poland; piotr.dziegiel@umw.edu.pl; 7Department of Human Biology, Faculty of Physiotherapy, Wroclaw University of Health and Sport Sciences, 51-612 Wroclaw, Poland; 8Department of Clinical Oncology, Wroclaw Medical University, 50-556 Wroclaw, Poland

**Keywords:** breast cancer, prolactin-inducible protein (PIP), gross cystic disease fluid protein 15 (GCDFP-15)

## Abstract

Prolactin-inducible protein (PIP), also referred to as gross cystic disease fluid protein 15 (GCDFP-15), has been a trending topic in recent years due to its potential role as a specific marker in breast cancer. PIP binds to aquaporin-5 (AQP5), CD4, actin, fibrinogen, β-tubulin, serum albumin, hydroxyapatite, zinc α2-glycoprotein, and the Fc fragment of IgGs, and the expression of PIP has been demonstrated to be modulated by various cytokines, including IL4/13, IL1, and IL6. PIP gene expression has been extensively studied due to its captivating nature. It is influenced by various factors, with androgens, progesterone, glucocorticosteroids, prolactin, and growth hormone enhancing its expression while estrogens suppress it. The regulatory mechanisms involve important proteins such as STAT5A, STAT5B, Runx2, and androgen receptor, which collaborate to enhance PIP gene transcription and protein production. The expression level of PIP in breast cancer is dependent on the tumor stage and subtype. Higher expression is observed in early-stage tumors of the luminal A subtype, while lower expression is associated with luminal B, basal-like, and triple-negative subtypes, which have a poorer prognosis. PIP expression is also correlated with apocrine differentiation, hormone receptor positivity, and longer metastasis-free survival. PIP plays a role in supporting the immune system’s antitumor response during the early stages of breast cancer development. However, as cancer progresses, the protective role of PIP may become less effective or diminished. In this work, we summarized the clinical significance of the PIP molecule in breast cancer and its potential role as a new candidate for cell-based therapies.

## 1. Introduction

In 1986, Haagensen et al. provided the initial description of this acidic protein, which was notably abundant in the gross cystic disease fluid of the breast [1,2]. Prolactin-inducible protein (PIP), also known as gross cystic disease fluid protein 15 (GCDFP-15), has emerged as a subject of significant interest in recent years due to its potential role as a specific marker in breast cancer.

This protein, encoded by the PIP gene, has been extensively studied for its diverse functions and intriguing characteristics. From its unique structural features to its involvement in regulatory pathways and clinical implications, PIP has captured the attention of researchers in the field. In the context of breast cancer, the expression of PIP holds clinical significance [3]. It varies depending on the tumor stage and subtype [4]. PIP’s role in response to chemotherapy highlights its potential as a predictive biomarker for treatment outcomes and is currently being investigated in clinical trials [5]. In this paper, we delve into the structural features, regulatory mechanisms, clinical implications, and potential therapeutic applications of PIP in breast cancer. We provide insights into its diverse functions, its impact on tumor progression, and its potential as a candidate for innovative cell-based therapies. Through a comprehensive exploration of the multifaceted roles of PIP, we aim to contribute to a deeper understanding of its significance in breast cancer research and treatment.

## 2. PIP: Structure, Expression, and Homeostatic Function

Prolactin-inducible protein (PIP), also referred to as gross cystic disease fluid protein 15 (GCDFP-15), extra-parotid glycoprotein (EP-GP), gp17 seminal actin-binding protein (SABP), or BRST2, is a fascinating protein with diverse functions and intriguing characteristics. Encoded by the PIP gene, this 146 amino acid pre-protein gives rise to a 118 amino acid secretory polypeptide, weighing approximately 13,506 kDa [6,7,8]. Notably, PIP exhibits microheterogeneity, as variations in its apparent molecular mass (14–20 kDa) have been observed across different sources [9,10]. The structural analysis of PIP conducted by Hassan et al. has shed light on its distinct features [11]. Researchers described structure of the ZAG-PIP complex and revealed a long interface between the two proteins, where the beta-structure of PIP aligns with the beta-structure of domain alpha3 of zinc 2-glycoprotein (ZAG). The protein is composed of seven parallel β-sheets and seven β-turns, with a lack of α-helix-type structures. Within its tertiary structure, PIP contains two disulfide bridges formed between cysteine residues at positions 37 and 63, as well as 61 and 95 [8,11]. Additionally, PIP displays N-glycosylation, which contributes to its apparent molecular mass variation and influences its isoelectric points [9,12]. A schematic representation of PIP as a 3D model is shown in Figure 1.

PIP is typically expressed in cells that possess apocrine properties, such as apocrine glands in various locations including the axilla, vulva, eyelid, ear canal, and seminal vesicle [13]. Additionally, PIP has been observed in serous cells of salivary glands, bronchial submucosal glands, and accessory lacrimal glands, which share some similarities with apocrine glands despite not being classified as typical apocrine epithelia [14,15]. As a secretory protein, PIP can be found in various bodily fluids such as seminal plasma, saliva, lacrimal fluid, tears, sweat gland secretion, amniotic fluid, and the blood of pregnant women [8,16]. Interestingly, it was also shown that PIP may bind to aquaporin-5 (AQP5), a significant protein involved in the production of saliva and lacrimal fluid [17,18,19]. This interaction results in the movement of AQP5 from its location in the cytoplasmic compartments to the apical membrane of acinar cells. Recently, PIP has been investigated as a potential biomarker for thyroid eye disease [20]. PIP is predominantly expressed in normal breast tissues but shows high abundance in metaplastic/hyperplastic apocrine epithelium of breast cysts and breast cyst fluid [21]. The expression of PIP is typically observed in breast tissue under physiological conditions. PIP is predominantly present within normal breast tissue, and its presence becomes lower to completely absent when the process of differentiation is lost due to the advancement of a tumor. The expression of PIP is closely connected with the differentiation of luminal epithelial cells in the mammary gland, a phenomenon that is induced by prolactin [22,23,24]. The existence of PIP under physiological and pathological conditions suggests that it is involved in many functional activities of cells.

Except for breast cancer, only a limited number of tumors, including prostate cancer and skin appendage carcinomas, exhibit the expression of PIP [5]. It was also found that PIP intensity is significantly higher in healthy subjects compared to keratoconus-diagnosed individuals, supporting its potential as a diagnostic marker for keratoconus [25]. Similar results were found in study conducted on patients with atopic dermatitis (AD), where PIP production by eccrine sweat glands was decreased [26]. Analysis of stratum corneum samples and sweat samples revealed lower levels of GCDFP-15 in AD compared to healthy individuals. Immunohistochemistry showed reduced GCDFP-15 expression in eccrine glands of AD skin specimens. Additionally, decreased expression of the cholinergic receptor M3 was observed in AD, potentially contributing to impaired sweating. These findings suggest that GCDFP-15 in stratum corneum could serve as a marker for dysregulated sweating in AD.

There is also some evidence from gene-based analyses about the PIP gene’s significance in association with anxious temperament and unipolar depression [27]. Figure 2 is a visual summary of the PIP expression profile in body tissues.

Various studies have unraveled the multifaceted roles of PIP in different biological contexts. PIP has been implicated in extracellular matrix degradation through its aspartyl protease activity, which stems from the presence of an aspartate residue at position 22 [28]. It binds to numerous proteins, including actin, fibrinogen, β-tubulin, serum albumin, hydroxyapatite, zinc α2-glycoprotein, and the Fc fragment of IgGs, among others [29,30]. Furthermore, PIP has been found to play a role in innate immunity by binding to bacteria and causing their aggregation [31,32]. This suggests that PIP contributes to the host’s defense against microbial infections, particularly in mucosal-type tissues and other potential points of pathogen entry.

PIP demonstrates the presence of CD4- and fibronectin-binding domains and PIP’s interactions with CD4 receptors present on T lymphocytes, macrophages, and spermatozoa have also garnered significant attention [33,34,35]. The binding affinity of PIP to CD4 receptors may vary depending on the source and physiological conditions [21]. Specifically, in PIP derived from human seminal plasma (gp17/SABP), the CD4-binding domain has been identified as two fragments [33,36,37]. The first fragment encompasses amino acids 1–35 at the N-terminus, while the second fragment encompasses amino acids 78–105. Additionally, the fibronectin-binding domain has been determined to consist of two peptides. The first peptide spans amino acids 109–118 at the C-terminus, while the second peptide covers amino acids 42–57.

PIP has been shown to inhibit the interaction of CD4 with the gp120 envelope glycoprotein of human immunodeficiency virus (HIV-1), suggesting its potential involvement in the pathogenesis of HIV-1 infection [35]. It is caused by inhibition of syncytium formation by cells expressing gp120 and CD4.

Moreover, PIP has demonstrated immunomodulatory functions by inhibiting T cell apoptosis induced by sequential gp120/CD4 and TCR/CD3 activation [38]. This antiapoptotic effect was associated with a moderate upregulation of Bcl-2 expression in treated cells. Studies have further linked PIP to cell-mediated immunity, highlighting its impact on the differentiation and production of cytokines by CD4-positive T cells and antigen-presenting cells [39,40]. The expression of PIP has been demonstrated to be modulated by various cytokines, including IL4/13, IL1, and IL6 [41]. In study by Blanchard et al., it was found that hPIP/GCDFP-15 glycoprotein, found predominantly in apocrine tissues, has elusive functions but has been implicated in breast cancer, metastasis, host defense processes, and T lymphocyte apoptosis [40]. The generation of PIP null mice (Pip−/−) allowed for investigations into its in vivo function, with no apparent developmental abnormalities, but histological findings indicated potential effects on lymph nodes, prostate lobes, and thymus. Microarray analysis revealed differential expression of genes related to cell death, survival, lipid metabolism, inflammation, immune disease, and cancer, suggesting an immunomodulatory role for PIP and offering insights into novel signaling pathways and regulatory networks associated with PIP. A recent study has demonstrated that PIP deficiency leads to reduced CD4(+) T cell numbers in peripheral lymphoid tissues and impairs CD4(+) Th1 cell differentiation [39]. PIP-deficient mice exhibited compromised CD4(+) T cell proliferation and IFN-γ production, indicating a potential impact on immune responses. Moreover, the susceptibility of PIP-deficient mice to Leishmania major infection highlights the importance of PIP in controlling lesion progression and parasite proliferation.

In addition to its immunological roles, PIP has been associated with fertility-related functions. In human seminal plasma, PIP acts as an IgG-reacting protein, binding to the Fc fragment of antisperm antibodies (ASAs) [42,43]. Reduced levels of PIP in the seminal fluid may be linked to infertility, especially in men with antisperm antibody (ASA), indicating the potential importance of PIP in countering the detrimental effects of ASA and supporting fertilization. The existence of PIP in human seminal fluid, coupled with its ability to suppress the immune response, indicates its potential role in counteracting the effects of ASA and its potential involvement in the breakdown of fibronectin during liquefaction, which could contribute to the fertilization process.

## 3. Regulatory Mechanisms of the PIP Gene

Expression of the PIP gene is a captivating process that has been the object of various research for years. The PIP gene is situated on the q34 locus of the long arm of chromosome 7 and spans a length of 7000 base pairs [44,45]. It comprises four exons and three introns. The expression of the PIP gene is influenced by multiple factors. Androgens, progesterone, glucocorticosteroids, prolactin, and growth hormone have been found to enhance the expression of the PIP gene, while estrogens have been shown to suppress its production [46,47,48]. The PIP-encoded protein has a predicted amino acid length of 146 aa, a mass of 16.6 kDa, and a secreted subcellular localization. Figure 3 is a visual representation that provides information about the antigen location on the target protein and features of the target protein.

Signal transducers and activators of transcription (STATs) are crucial contributors to the regulatory processes governing the PIP gene. Within the array of STATs, STAT5A, and STAT5B are the key participants in overseeing PIP regulation. These two closely related proteins exhibit a remarkable 96% sequence similarity and stem from distinct yet closely linked genes [49].

When prolactin binds to the PRL receptor, it triggers phosphorylation on Tyr694 in Stat5A and Tyr699 in Stat5B. This phosphorylation process can occur through Janus kinase (JAK), which is associated with the receptor. This process confers them the ability to undergo dimerization, enabling their entry into the nucleus [50]. In the nucleus, they bind to the STAT5-responsive element situated on the promoter of the PIP gene, and in collaboration with an activated androgen receptor bound to androgen-responsive elements on the PIP promoter, they work together to enhance the transcription of the PIP gene [21]. The interaction between Stat5 and the androgen receptor was demonstrated for the first time in the scientific study authored by Carsol et al. [50].

Figure 4 shows schematic representation of one of the regulatory mechanisms of PIP.

Another way of increasing PIP translation is through a mechanism involving Runx2. Runx2 (also known as CBFA1) is a member of the Runt family of transcription factors. It is widely recognized as a pivotal controller that governs the process of bone development [51]. Its high expression is a poor prognostic marker in osteosarcoma [52].

In the article by Inman et al., it was shown that the Runx2 protein is expressed in several mammary epithelial cell lines and in primary mammary epithelial cells. Runx2 protein was detected by Western blot analysis of nuclear extracts. The study found that Runx2 expression is not restricted to metastatic human breast cancer cell lines (MDA-MB-231) but is also clearly expressed in non-metastatic human breast cancer cells (MCF-7), non-tumorigenic (HC11) cells, and primary mammary epithelial cells [53].

In the research conducted by Selvamurugan et al., Western blot analysis revealed the presence of the Runx2 protein within mammary epithelial cells of the MDA-MB-231 cell line [54].

In the investigation conducted by Onodera and co-authors, a total of one hundred and thirty-seven formalin-fixed and paraffin-embedded breast cancer specimens were utilized for an immunohistochemical analysis. Immunoreactivity was assessed using the labeling index (LI). The immunoreactivity of Runx2 was observed in both carcinoma cells and stromal cells, as well as non-pathological ductal cells. Notably, the nuclear labeling index (LI) of Runx2 in carcinoma cells exhibited associations with the clinical stage, histological grade, and HER2 status of the patients under examination. Furthermore, among patients without distant metastasis, those with a high Runx2 labeling index (LI) demonstrated a significantly inferior clinical outcome compared to those with a low LI [55].

The previously established knowledge indicated that androgens induce PIP expression in different breast cancer cell lines such as T47D, ZR-75, and MDA-MB-453 [7,56]. In a study led by Baniwa et al., a novel mechanism by which androgens stimulate PIP through Runx2 was elucidated for the first time [57]. The investigation encompassed human breast cancer T47D cells and prostate cancer C4-2B cells, highlighting the pivotal roles of AR, Runx2, and dihydrotestosterone (DHT) in this pathway.

Notably, both AR and Runx2 transcription factors exhibit specific sequences preceding the PIP transcription start site. In analysis by Sandelin et al., four distinct regions were identified and designated as R-I (−0.9 kb), R-II (−2.4 kb), R-III (−9.4 kb), and R-IV (−11 kb) [58]. Regions I and II serve as binding sites for Runx2, while regions III and IV act as binding sites for both Runx2 and AR.

Upon activation by 5α-dihydrotestosterone of AR, AR and Runx2 initiate the process. They subsequently bind to the enhancer element of the PIP promoter. By engaging in both physical and functional interactions, these factors synergistically work together to greatly amplify the expression of the PIP gene [59,60]. PIP is also involved in facilitating the translocation of the androgen receptor to the nucleus and stimulation of androgen-dependent genes which creates a positive feedback loop [57].

Figure 5 summarizes the process of the PIP gene activation through AR and Runx2.

Another regulating mechanism of the PIP gene was found in a subtype of ER-negative, AR-positive breast cancer called molecular apocrine. A positive feedback loop involving PIP and ERK, Akt/PkB signaling was discovered by A. Naderi et al. [29].

PIP possesses an aspartic-type protease activity that specifically targets fibronectin, breaking it down into smaller protein fragments. These fragments, in turn, have a capability to activate β1-integrins—cell surface receptors. Once activated, β1-integrins trigger a cascade of events involving binding ILK1 and the plasma membrane receptor ErbB2. ILK1 initiates the induction of ERK and Akt (also known as Akt/PKB or PI3K-Akt) signaling pathways [61,62,63]. Notably, ERK itself is activated by ErbB2. Both ERK and Akt play vital roles in phosphorylating the RSK and MSK kinase families, leading to the activation of CREB1 by phosphorylation by those kinases [64,65]. CREB1, acting as a transcription factor for the PIP gene, enhances its transcription, leading to an increased production of PIP.

The regulatory mechanism involving PIP exhibits several positive feedback loops. Firstly, PIP plays an active role in degrading fibronectin, which is crucial to activate β1-integrins, establishing a positive feedback loop. Furthermore, CREB1 acts as a transcription factor for AR, which, as mentioned earlier, enhances the production of PIP through pathways involving Stat5 and Runx2. Additionally, AR contributes to a positive feedback loop by inducing the expression of ErbB2, thereby activating the ERK-CREB1 axis [66]. These interconnected feedback loops play crucial roles in regulating PIP expression and its downstream effects, highlighting the complexity of the regulatory network involved. The most important elements of this process are depicted in Figure 6.

## 4. Prognostic Significance of PIP Expression in Breast Cancer

Some studies indicate that prolactin-inducible protein (PIP) is expressed to varying degrees in more than 90% of breast cancers (BCs) [67,68]. However, in more recent publications, the information regarding expression indicates slightly lower levels of PIP. In a study conducted by Darb-Esfahani et al., only 39.7% of breast tumors were PIP positive [5]. Similarly, a recent study by Shiran et al. observed PIP expression in 20 out of 60 breast cancer patients (33.3%) [69]. In another paper, histopathologic testing on apocrine carcinoma of the breast revealed that 60% of tumors were PIP positive [70].

The expression level of PIP is dependent on the stage of the tumor, as higher expression is observed in G1 and G2 than in G3 [4]. Interestingly, the expression levels also vary among subtypes. They are highest in the luminal A subtype, followed by the HER2+ and normal-like subtypes. In contrast, the lowest expression is observed in the luminal B and basal-like subtypes, with an even lower expression noted in the triple-negative subtype with a poor prognosis. A study by Gangadharan et al. revealed a substantial downregulation of PIP transcription in cancer samples when compared to normal breast tissue. This downregulation was particularly pronounced in advanced-grade tumors, where mRNA levels were significantly decreased by 93-fold compared to lower-grade tumors [71]. Interestingly, the downregulation of PIP transcription was also evident in early-stage samples of both triple-negative and hormone receptor-positive tumors, suggesting that PIP may play a role in the early stages of breast cancer development. Molecular apocrine tumors have higher levels of PIP expression among ER-negative breast tumors. Among ER-negative tumors, PIP expression is 57% in apocrine and 16% in AR- [72]. In breast tumors, PIP expression is correlated with apocrine differentiation, recurrence-free survival, and hormone receptor and HER2 positivity [71]. In addition, higher PIP expression is characterized by significantly longer metastasis-free survival [4]. PR+ breast tumors show significantly higher levels of PIP expression than PR- cases. There was also a negative correlation between PIP expression and proliferation marker Ki-67 and no association with cytokeratins 5, 6. Notably, PIP-positive cells exhibit heightened expression levels of genes associated with antiproliferative and proapoptotic effects in comparison to their PIP-negative counterparts within the breast cancer cell population [73]. Remarkably, PIP has been unveiled as a key participant in intracellular signaling cascades, exerting its influence either directly or indirectly upon an array of molecular actors. These include focal adhesion kinase (FAK), ephrin B3 (EphB3), tyrosine kinase of the Src family (FYN), hemopoietic cell kinase (HCK), and serine/threonine kinases AKT, ERK1/2, c-jun N-terminal kinase (JNK1), along with CREB1 and integrin-activated signaling pathways [57,74]. Taken together, several studies confirm that high PIP expression is significantly associated with BC having a good prognosis [5,75].

Research using PIP KO mice has shown that the absence of PIP can compromise the Th1 immune response, making individuals more susceptible to intracellular pathogens that are normally controlled by cell-mediated immunity [39]. This indicates that PIP may have a protective role in breast cancer. This finding aligns with previous observations that a strong Th1 immune response is beneficial in breast cancer, as patients whose tumors elicit a robust Th1 response tend to have a better prognosis compared to those whose tumors exhibit a T-helper 2 (Th2) response [76]. To date, studies have demonstrated that breast cancer patients with larger tumors exhibit a decreased response of Th1 cytokines (such as IL-12 and IFN-g) [77]. This implies that during the initial stages of breast cancer development, PIP may play a role in supporting the immune system, boosting its ability to mount a strong antitumor response against cancer cells that are recognized as foreign. However, as cancer progresses, this protective role of PIP may become less effective or diminished, which could explain the lower expression of PIP observed in tumors with a less favorable prognosis [78].

Clinically relevant low or no PIP expression is associated with poorer response of breast cancer cells to chemotherapy. The study by Urbaniak et al. revealed BC cells with high expression of PIP were more sensitive to the cytotoxic effects of chemotherapeutic agents, including doxorubicin (DOX), 4-hydroxycyclophosphamide (4-HC), and paclitaxel (PAX), compared to cells with no expression of PIP. Conversely, BC cells with knockdown of the PIP gene exhibited reduced sensitivity to DOX and 4-HC. These results support studies indicating that high PIP expression in BC tissues is associated with a positive response to chemotherapy [3]. This effect is linked to the increased expression of certain proapoptotic genes (CRADD, DAPK1, FASLG, CD40, and BNIP2). The proapoptotic action of PIP is facilitated through the release of PIP into the surrounding environment, potentially involving a specific surface receptor. Notably, a significant correlation between high PIP expression and a favorable response to anticancer drugs was observed, suggesting that this glycoprotein could serve as a prognostic marker for assessing the effectiveness of adjuvant chemotherapy. These findings are consistent with a study by Jablonska et al. which suggests that high PIP expression is correlated with a better response to adjuvant therapy, implying that PIP may serve as a potential biomarker for predicting chemotherapy response and clinical outcomes in BC patients [4]. Interestingly, in breast mucinous carcinoma, GCDFP-15 (also known as PIP) expression is strongly associated with older age, irrespective of the carcinoma subtype or other immunohistochemical markers [79]. Moreover, apocrine markers (GCDFP-15/AR) were more prevalent in older patients and characterized mucinous carcinoma in this age group and some studies suggest that the biological characteristics of mucinous carcinoma in older patients are apocrine-like features rather than neuroendocrine features [80,81]. Other immunohistochemical markers, such as CGA, SYP, CD56, AR, ER, PgR, HER2, and Ki-67, did not show consistent associations with age or carcinoma type [79]. These conclusions highlight the clinicopathological and immunohistochemical characteristics of breast mucinous carcinoma in older and younger patients. The findings emphasize the relevance of GCDFP-15 and apocrine markers in characterizing mucinous carcinoma in older patients.

It is worth noting that among other breast cancer markers, such as mammaglobin or GATA-4, PIP is distinguished from the others by its high specificity, which makes it extremely valuable [82].

In the GeparTrio trial (NCT00544765), a phase III randomized neoadjuvant study for breast cancer treatment, immunohistochemistry was used to examine the expression of GCDFP-15 (known as PIP) in 602 pre-therapeutic core biopsies obtained from patients [5]. Findings indicate that the expression of GCDFP-15 is observed in various breast cancer subtypes, with higher prevalence in hormone receptor (HR)-positive and HER2-positive tumors, while its expression is relatively low in the triple-negative breast cancer (TNBC) subtype. Thus, results suggest that GCDFP-15 can serve as a diagnostic marker for mammary differentiation in HR- and HER2-positive tumors but has reduced sensitivity in TNBC. Although GCDFP-15 expression is associated with favorable clinicopathological factors, it is not an independent prognostic factor for overall survival (OS) or disease-free survival (DFS). Its prognostic impact is likely correlated with other factors such as HR expression, nodal stage, and tumor grade. Interestingly, GCDFP-15 expression is not predictive of response to anthracycline/taxane-based neoadjuvant chemotherapy (NACT). It does not serve as a reliable predictor for the tumor’s response to NACT within specific breast cancer subtypes.

Another study about the clinical significance of neuroendocrine (NE) carcinoma of the breast revealed that the expression of GCDFP-15 in NE breast tumors is related to hormone dependency [83]. Specifically, the presence of androgen receptors (ARs) correlates with the expression of GCDFP-15. The short-term follow-up analysis did not show a significant improvement in prognosis for patients with NE/apocrine tumors expressing GCDFP-15. However, in the long term (five years after surgery), the association with apocrine differentiation appeared to enhance the overall survival of patients with NE tumors.

A study conducted on 165 patients revealed that GCDFP-15 and mammaglobin are both widely expressed markers in primary breast cancer, with cytoplasmic expression observed in 73.3% and 72.1% of invasive breast carcinomas, respectively [84]. Both GCDFP-15 and mammaglobin expression are significantly associated with lower tumor grading, indicating a potential link to less aggressive tumor biology. GCDFP-15 negativity is significantly associated with shortened disease-free survival times in patients with breast cancer. This prognostic value of GCDFP-15 was validated in both univariate and multivariate analyses. These findings suggest that GCDFP-15 and mammaglobin together are valuable markers for breast cancer diagnosis, with GCDFP-15 exhibiting prognostic significance in predicting disease-free survival.

Interestingly, results from a study conducted on a group of 135 patients indicate that elevated blood levels of GCDFP-15 in women with active breast gross cystic disease may indicate an increased risk of developing breast cancer [85]. However, it is worth noting that post-menopausal women with active breast gross cystic disease generally have lower GCDFP-15 plasma levels compared to those with clinically active disease. This study also evaluated the relationship between blood levels of GCDFP-15 in women with active breast gross cystic disease confirmed by cyst aspiration and the risk of developing breast cancer. The risk of breast cancer was higher in women with elevated GCDFP-15 plasma levels, especially among those with 10 or more total aspirated cysts.

It was shown that GCDFP-15 is a highly specific and sensitive marker even for breast metastases and can be used to distinguish breast tissue origin from others [86]. Breast cancer metastasis to the stomach is relatively rare but can be mistaken for primary stomach cancer, underscoring the importance of accurate diagnosis to determine appropriate treatment. Case studies described by Park et al. demonstrate the effectiveness of GCDFP-15 immunohistochemical staining in confirming the diagnosis of gastric metastasis from breast cancer. Thus, this study highlights the value of using GCDFP-15 as a specific immunocytochemical marker to aid in the diagnosis of gastric metastasis from breast carcinoma, enabling appropriate treatment decisions. Interestingly, studies conducted on mouse models confirmed that PIP modulates antitumor immune responses and metastasis in triple-negative breast cancer [68]. Research on 4T1 and E0771 mouse BC cell lines with stable PIP expression showed minimal impact on in vitro behaviors. In vivo experiments using the 4T1 syngeneic mouse model revealed PIP-expressing tumors exhibited delayed onset and reduced growth, tied to increased natural killer cells and decreased type 2 T-helper cells. Interestingly, PIP expression was paradoxically associated with elevated lung metastatic colonies, possibly due to upregulated metastasis-related genes in PIP-expressing cells. This suggests PIP’s dual role in BC, affecting both antitumor immunity and metastasis. In a distinct investigation encompassing a cohort of 111 patients, the presence of antibodies targeting GCDFP-15/gp17 was assessed, revealing their detection in both individuals afflicted with mammary carcinoma and those with benign breast conditions [87]. Among patients with benign breast conditions, a minority (2.6%) exhibited detectable levels of antibodies against GCDFP-15/gp17, while a slightly higher proportion (5.5%) of patients with malignant mammary gland tumors displayed statistically significant antibody concentrations.

Of particular note, the highest concentrations of circulating anti-GCDFP-15/gp17 antibodies were observed in patients diagnosed with highly malignant ductal or lobular breast carcinoma and in those with dysplasia. Intriguingly, no discernible correlations emerged between the presence of these circulating antibodies and factors such as tumor size or patient age. However, a bimodal relationship did manifest in connection with the percentage of invaded lymph nodes. The identified antibodies primarily belonged to the IgM and IgG isotypes, implying the involvement of a T-cell-mediated immune response. These findings collectively suggest the potential utility of the anti-GCDFP-15/gp17 immune response as an investigative tool for unraveling specific facets of breast disease progression. Furthermore, this study advances the notion that GCDFP-15/gp17 may merit exploration as an antigenic candidate in the pursuit of prospective antitumor vaccination strategies for breast cancer.

## 5. Exploring the Impact of PIP in Cell-Based Therapies: Insights from Mouse Models and Tumor Growth Studies

Cell-based therapies have emerged as cutting-edge treatments that have shown remarkable therapeutic achievements [88]. These therapies hold great potential for addressing a wide range of currently difficult-to-treat diseases, thanks to their highly powerful mechanisms of action.

To investigate the role of PIP in vivo, a PIP null mouse model (Pip−/− mouse) was generated in study by Blanchard et al. [40]. The Pip−/− mice developed normally, showing no overt differences in behavior or gross morphology, and they were fertile. Microarray analysis of the submandibular gland in Pip−/− mice showed differential expression of genes associated with cell death and survival, lipid metabolism, inflammation, immune disease, and cancer. Furthermore, PIP deficiency is linked to impaired signaling processes within macrophages and DCs, resulting in reduced production of proinflammatory cytokines. These findings suggest that PIP may have an immunomodulatory role in vivo and provide insights into potential signaling pathways and regulatory networks involving PIP. Another study provided evidence that mice lacking PIP exhibit a hindered ability to differentiate CD4+ Th1 cells, indicating the essential role of PIP in promoting effective cell-mediated immune response [39].

A recent study by Edechi et al. found that, in immunocompetent mice, the presence of PIP resulted in delayed tumor onset and reduced tumor growth compared to control tumors [68]. However, in immunodeficient mice lacking key immune cells, there was no difference in tumor growth between PIP-expressing cells and control cells. This suggested that the effect of PIP on primary tumor growth was at least partially mediated by the immune response. Further analysis revealed that PIP expression altered the composition of immune cells within the tumors. PIP-expressing tumors had higher percentages of natural killer (NK) cells and dendritic cells (DCs), which are known to promote antitumor immune responses. Conversely, there were lower percentages of Th2 cells, which are protumorigenic. These findings suggested that PIP expression was modifying the antitumor immune response. To confirm the role of immune cells in the observed tumor growth effects, PIP-expressing cells were injected into immunodeficient mice lacking NK, T, and B cells. In these mice, there was no difference in tumor growth between PIP-expressing cells and control cells, indicating that the immune response mediated the reduced tumor growth in immunocompetent mice. The study also investigated the sensitivity of PIP-expressing cells to commonly used chemotherapeutic agents for breast cancer treatment. PIP expression did not directly affect the sensitivity of the cells to these drugs. Furthermore, PIP did not influence the sensitivity of the cells to tamoxifen, a selective estrogen receptor modulator.

Further, in a study by Terceiro et al., a PIP knockout (KO) mouse model was generated using the CRISPR/Cas9 system by injecting Cas9 protein and guiding RNAs targeting specific regions of the PIP gene into mouse embryos [19]. The PIP KO mouse model was generated in the C57Bl/6J mouse strain, which has low genetic variation, allowing for more controlled and homogenous experimental conditions. The authors suggest that the model can be used to investigate the role of PIP in breast cancer, keratoconus, Sjögren’s syndrome, and other diseases attributed to the deregulation of the PIP gene.

## 6. Conclusions

Prolactin-induced protein (PIP) is considered as a highly promising prognostic factor in breast cancer treatment. PIP is a highly specific and sensitive marker for breast tissue origin, even in metastatic sites. The presence of antibodies against PIP/GCDFP-15 has been detected in patients with both benign and malignant breast conditions, suggesting a potential immune response and its relevance in investigating breast disease progression and vaccination strategies. The expression of PIP/GCDFP-15 in breast cancer is associated with hormone receptor-positive and HER2-positive tumors, while its expression is relatively low in triple-negative breast cancer. PIP/GCDFP-15 can serve as a diagnostic marker for mammary differentiation in HR- and HER2-positive tumors but has reduced sensitivity in TNBC. In this paper, we highlight the clinical significance of PIP/GCDFP-15 as a diagnostic and prognostic marker in breast cancer, its association with specific tumor subtypes, and its potential implications in immunotherapy and targeted treatments. This manuscript provides a comprehensive review of the current knowledge surrounding the structure, function, and regulation of PIP expression, highlighting its clinical significance.

## Figures and Tables

**Figure 1 cells-12-02252-f001:**
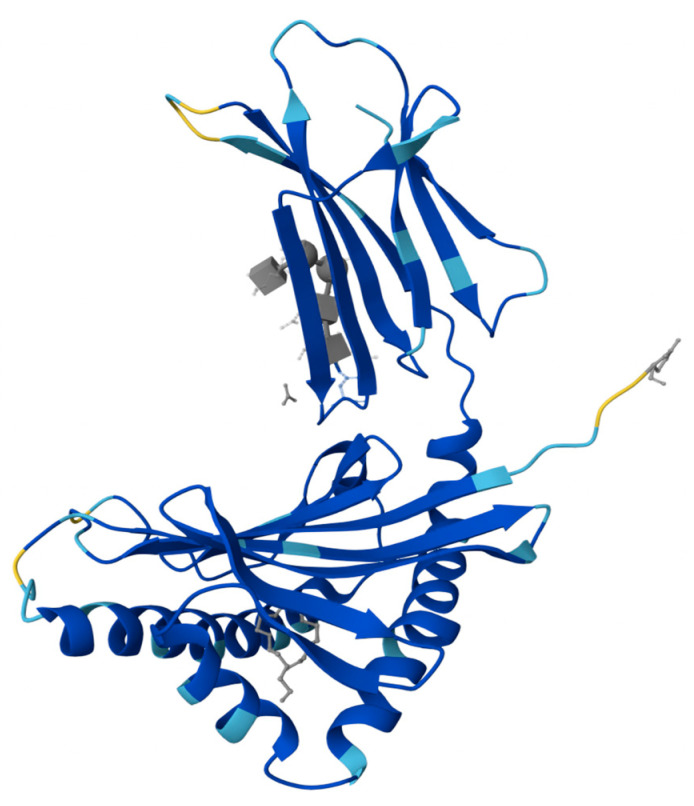
Crystal structure of the complex formed between zinc 2-glycoprotein (ZAG) and prolactin-inducible protein (PIP) [11].

**Figure 2 cells-12-02252-f002:**
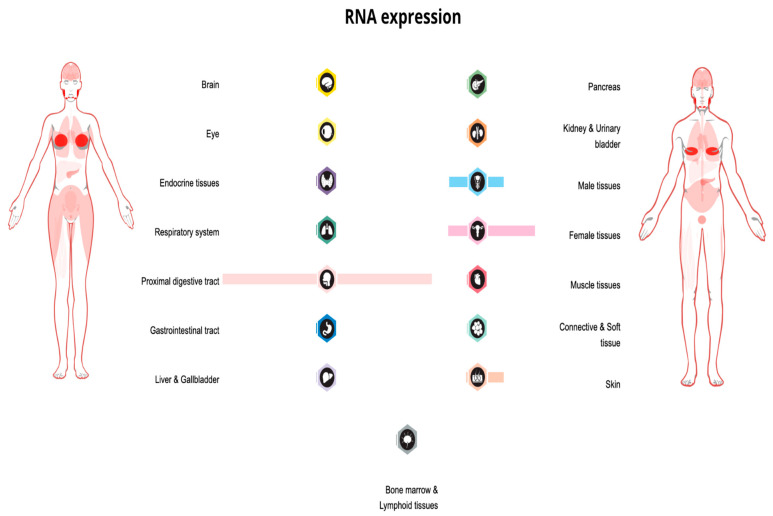
PIP expression overview in tissues of the human body, visualization shows RNA-seq data generated by The Cancer Genome Atlas (TCGA) (image credit: Human Protein Atlas www.proteinatlas.org. Image available at the following URL: https://v21.proteinatlas.org/ENSG00000159763-PIP/tissue, accessed on 9 July 2023).

**Figure 3 cells-12-02252-f003:**
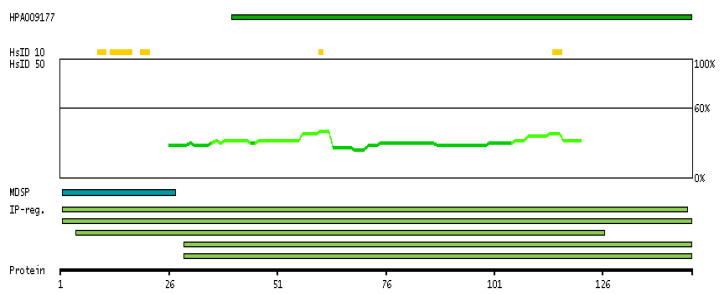
The antigen location on the target protein(s) and the features of the target protein. At the top, the position of the antigen (identified by the corresponding HPA identifier) is shown as a green bar. Below the antigens, the maximum percent sequence identity of the protein to all other proteins from other human genes is displayed, using a sliding window of 10 aa residues (HsID 10) or 50 aa residues (HsID 50). The region with the lowest possible identity is always selected for antigen design, with a maximum identity of 60% allowed for designing a single-target antigen. The curve in blue displays the predicted antigenicity, i.e., the tendency for different regions of the protein to generate an immune response, with peak regions being predicted to be more antigenic. The curve shows average values based on a sliding window approach using an in-house propensity scale. If a signal peptide predicted by a majority of the signal peptide predictors SPOCTOPUS, SignalP 4.0, and Phobius (turquoise) is predicted by MDM, it is displayed. Low-complexity regions are shown in yellow and InterPro regions in green. Common (purple) and unique (grey) regions between different splice variants of the gene are also displayed, and at the bottom of the protein view is the protein scale. (Image credit: Human Protein Atlas www.proteinatlas.org. Image available at the following URL: https://v21.proteinatlas.org/ENSG00000159763-PIP#gene_information, accessed on 9 July 2023).

**Figure 4 cells-12-02252-f004:**
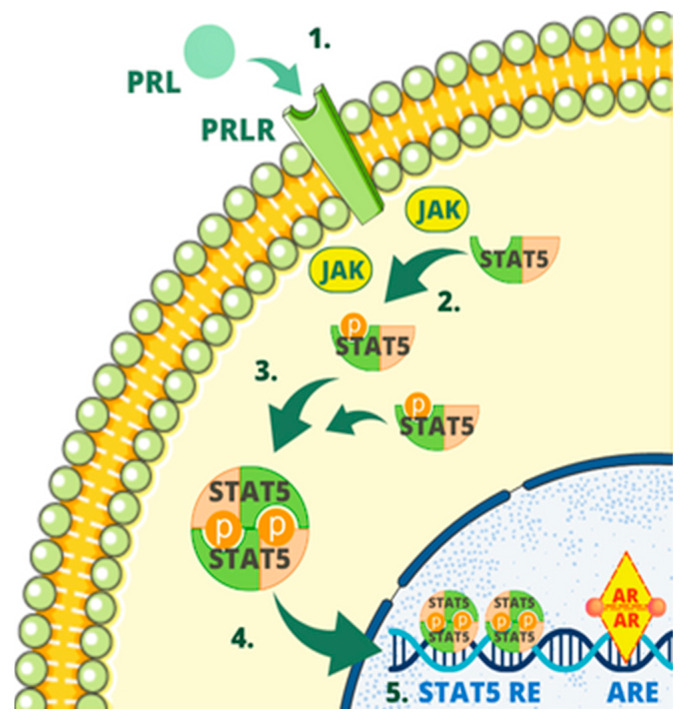
One of the regulatory mechanisms of PIP. The mechanism involves several steps as follows: (1) Prolactin (PRL) binding to prolactin receptor (PRLR); (2) phosphorylation in STAT5; (3) dimerization of STAT5; (4) entering the nucleus by STAT5; (5) binding to the STAT5-responsive element (STAT5 RE) by STAT5, binding by the androgen receptor (AR) to androgen-responsive elements (AREs), enhancing PIP expression of PIP gene.

**Figure 5 cells-12-02252-f005:**
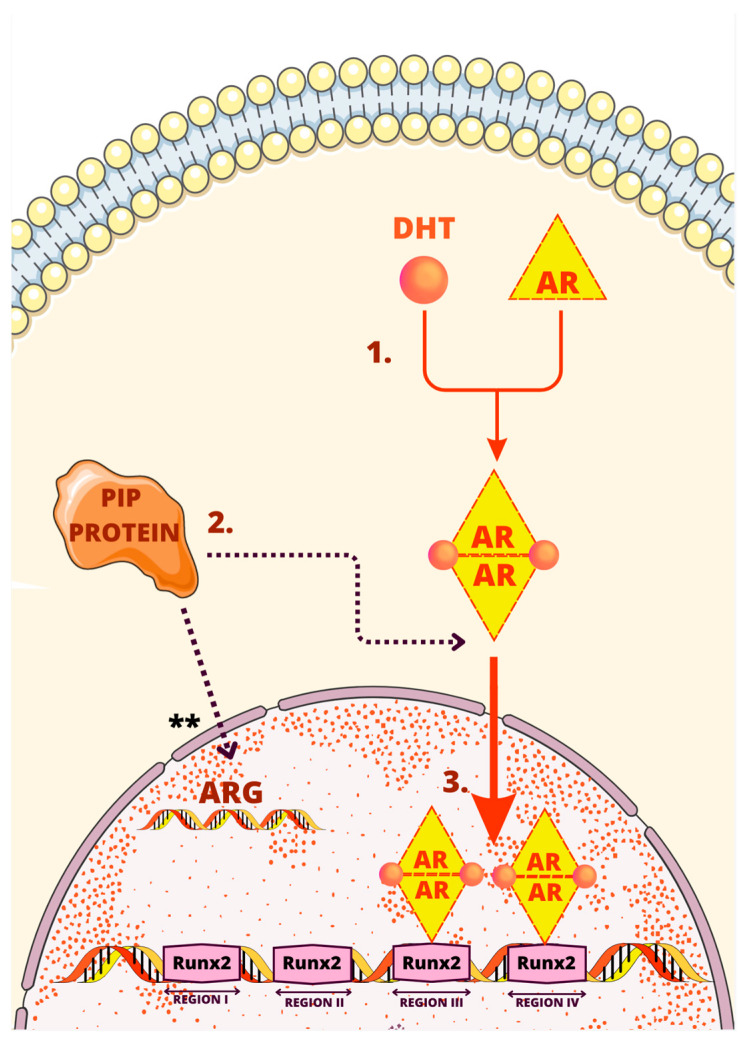
One of the regulatory mechanisms of PIP. The mechanism involves several steps as follows: (1) Binding 5α-dihydrotestosterone (DHT) to the androgen receptor (AR) inducing dimerization of the AR; (2) AR entering nucleus facilitated by PIP; (3) binding to enhancer element of the PIP promoter by androgen receptor and Runx2 increasing expression of the PIP gene. ** Stimulation of androgen-dependent genes by PIP.

**Figure 6 cells-12-02252-f006:**
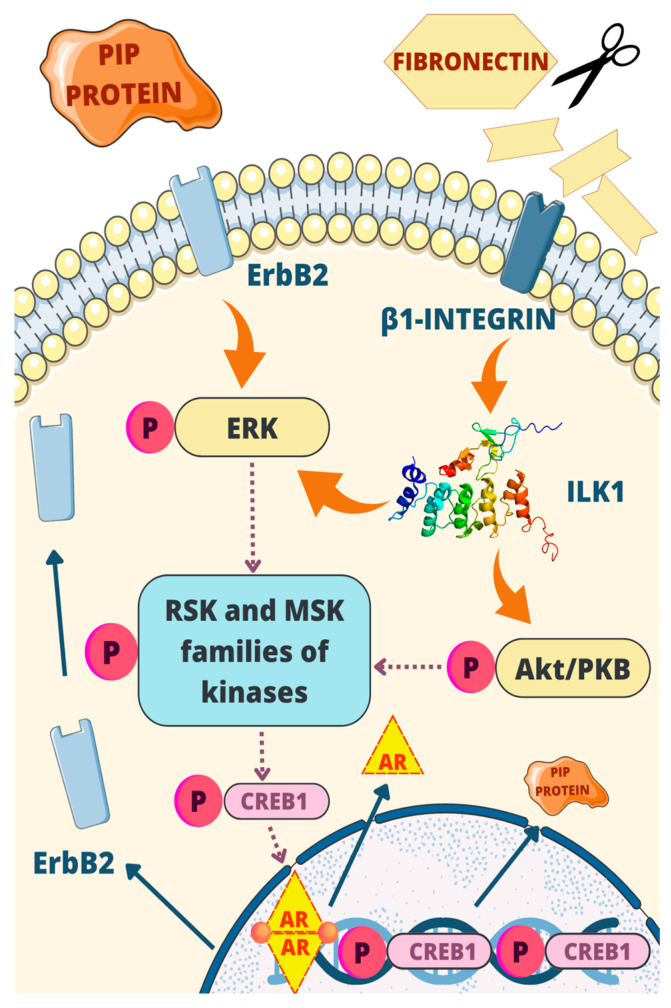
One of the regulatory mechanisms of PIP: PIP possesses aspartic-type protease activity, targeting fibronectin and activating β1-integrin. This triggers a cascade involving ILK1, ErbB2, ERK, and Akt signaling pathways. ERK and Akt phosphorylate RSK and MSK kinases, leading to CREB1 activation and increased PIP gene transcription. Positive feedback loops involve PIP degradation of fibronectin, CREB1 transcriptional regulation, and AR-mediated induction of ErbB2.

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
