# Peer review of "Prognostic Role of Prolactin-Induced Protein (PIP) in Breast Cancer"

_cells, 2023, doi:10.3390/cells12182252_

Round 1

Reviewer 1 Report

Major critiques:

1)     PIP is mainly expressed in normal breast and its expression is diminished to zero with loss of differentiation associated with tumor progression. Although the authors cite many papers supporting that, they do not connect PIP expression directly with differentiation of mammary luminal epithelial cells (it is prolactin-induced) but rather with vaguely described immunomodulatory function and other still putative roles.  This creates some confusion about what this protein actually does, and whether or not its expression important for breast cancer (BCa) growth and metastasis. One needs to acknowledge that some proteins ceased to be expressed in de-differentiated tumor cells and this might be not necessarily impactful for key clinical parameters and/or disease outcome.

2)     Related to Point#1, PIP is described as being modulated by immunosuppressive Th2 cytokines IL-4/IL-13 (line 117) and having STAT5 binding site on the promoter (line 181), but at the same time it promotes stimulatory Th1 response and correlates with better prognosis. These statements and conclusions appear to contradict each other. Note that STAT5 can be activated by prolactin but also by Th2 cytokines.

3)     Potentially anti-tumor beneficial role of PIP in BC is not consistent with its detection in metastatic lesions. Given the fact that PIP is downregulated with tumor progression, it does not appear to be a highly reliable marker for metastases. The authors need to put this information in some perspective how two contradictory roles are mediated by the same protein.

4)     RUNX2 is a myeloid-macrophage specific transcription factor. It is not expressed in mammary or other epithelial cells. The RUNX2 dependent mechanism of PIP regulation described on pages 5-6 is not relevant to its role in breast carcinoma epithelial cells. In context of tumors, it is only relevant to osteosarcoma (line 197) because osteoclasts are bone macrophages that do express RUNX2. The presented information implies that RUNX2 regulates PIP expression in BC cells which is not very likely. Unless authors can cite a reference that shows protein expression of RUNX2 (not mRNA) in human breast cancer, the described mechanism should be cited with some caution.

5)     Reference 59 is cited as the published evidence of PIP expression in 90% of BC but this paper from 1999 does not contain any images of stained tumors, only PCR data. The authors need to determine the quality and validity of cited evidence before distributing it for further dissemination in a review article. Other studies determined much lower percentage of PIP positive tumor (e.g., 39% in reference 66), and Human Protein Atlas shows extremely limited expression in 1-2 well-differentiated tumors with zero expression in multiple de-differentiated cancers. It is entirely possible that loss of differentiation is a cause for poor prognosis, not downregulation or reduced expression of PIP. The statement on lines 277-278 might be similarly interpreted – tumors that still retain tissue-specific architecture might have better prognosis, which may or not have any relevance to expression of PIP.

6)     Conclusions state that PIP is a highly promising therapeutic factor but no published evidence is cited to support this statement. The cited papers noted only correlation (not causation) between lower PIP and higher grades, which could be explained by coincident downregulation of PIP in de-differentiating tumor cells. That does not mean that PIP inhibits progression of malignancy, and this certainly does not mean that upregulation of PIP can prevent or suppress development of malignancy (no evidence for a therapeutic effect). It is also unlikely to play an anti-tumor role and at the same time be a marker of metastasis. There is also very little evidence that PIP can be used for anti-tumor vaccination strategy, particularly in the light of its low expression in tumors other than early-staged and highly differentiated BC. There is no doubt that PIP is an interesting protein but conclusions should be a little bit more objective with regard to its actual clinical significance in BCa.

Minor critiques:

1)     Line 48 – define ZAG

2)     Line 80 – define or edit “suggestive significance”

3)     Line 199 – should be MDA-MB-453

4)     Line 203 – the sentence needs to be edited for proper grammar

5)     Line 264 – should be ER-negative

6)     Line 270 – should be cytokeratins 5, 6

7)     Line 366 – should be breast cancer

8)     Last name of authors can be included but no need for the first name (line 200) and/or first initial (line 224)

9)     Define “molecular apocrine” breast cancer

quality is very good, only a few minor suggestions 

Author Response

Dear Reviewer,

We sincerely appreciate your thorough review and insightful comments on our manuscript. Your feedback has been immensely valuable in refining our work. We have carefully considered each of your points and made the necessary revisions to address your concerns. 

Below are the original comments and our corresponding responses:

1)     PIP is mainly expressed in normal breast and its expression is diminished to zero with loss of differentiation associated with tumor progression. Although the authors cite many papers supporting that, they do not connect PIP expression directly with differentiation of mammary luminal epithelial cells (it is prolactin-induced) but rather with vaguely described immunomodulatory function and other still putative roles.  This creates some confusion about what this protein actually does, and whether or not its expression important for breast cancer (BCa) growth and metastasis. One needs to acknowledge that some proteins ceased to be expressed in de-differentiated tumor cells and this might be not necessarily impactful for key clinical parameters and/or disease outcome.

Thank you for your valuable comment. We have revised the manuscript and added more about the role of PIP in the physiological state in the breast: “PIP is predominantly expressed in normal breast tissues but shows high abundance in metaplastic/hyperplastic apocrine epithelium of breast cysts and breast cyst fluid [21]. The expression of PIP is typically observed in breast tissue under physiological conditions. PIP is predominantly present within normal breast tissue, and its presence becomes lower to completely absent when the process of differentiation is lost due to the advancement of a tumor. The expression of PIP is closely connected with the differentiation of luminal epithelial cells in the mammary gland, a phenomenon that is induced by prolactin [22–24]. The existence of PIP under physiological and pathological conditions suggests that it is involved in many functional activities of cells.  “

2)     Related to Point#1, PIP is described as being modulated by immunosuppressive Th2 cytokines IL-4/IL-13 (line 117) and having STAT5 binding site on the promoter (line 181), but at the same time it promotes stimulatory Th1 response and correlates with better prognosis. These statements and conclusions appear to contradict each other. Note that STAT5 can be activated by prolactin but also by Th2 cytokines.

Regarding Point#1, we acknowledge the apparent contradiction in the functions and effects of PIP as outlined in the manuscript. Indeed, the immunomodulatory roles of PIP seem to be multifaceted and context-dependent. PIP's modulation by immunosuppressive Th2 cytokines (IL-4/IL-13) and its interaction with STAT5 on the promoter suggest a complex interplay between immune responses and regulatory mechanisms. However, as you've astutely pointed out, it's intriguing that PIP also promotes stimulatory Th1 responses and correlates with better prognosis. This duality warrants further investigation and may be attributed to intricate cross-talk between different signaling pathways, cell types, and microenvironmental factors. The observation that STAT5 can be activated by both prolactin and Th2 cytokines adds an additional layer of complexity to these interactions.The coexistence of seemingly opposing functions within PIP's immunomodulatory roles may be attributed to the dynamic nature of the immune response and the versatility of PIP's actions. It is possible that PIP's effects are contingent upon the specific immune context and the overall physiological state of the system. These complexities merit careful consideration and exploration in future studies. The manuscript aims to present a comprehensive overview of the existing literature on PIP's roles and regulatory mechanisms, acknowledging the nuances and contradictions within the field. 

3)     Potentially anti-tumor beneficial role of PIP in BC is not consistent with its detection in metastatic lesions. Given the fact that PIP is downregulated with tumor progression, it does not appear to be a highly reliable marker for metastases. The authors need to put this information in some perspective how two contradictory roles are mediated by the same protein.

Thank you for your comment.  We have revised the manuscript and added more information: “It was shown that GCDFP-15 is a highly specific and sensitive marker even for breast metastases and can be used to distinguish breast tissue origin from others [88]. As breast cancer metastasis to the stomach is relatively rare, but can be mistaken for primary stomach cancer, underscoring the importance of accurate diagnosis to determine appropriate treatment. Case studies described by Park et al. demonstrate the effectiveness of GCDFP-15 immunohistochemical staining in confirming the diagnosis of gastric metastasis from breast cancer. Thus, this study highlights the value of using GCDFP-15 as a specific immunocytochemical marker to aid in the diagnosis of gastric metastasis from breast carcinoma, enabling appropriate treatment decisions. Interestingly, studies conducted on mouse models confirmed that PIP modulates antitumor immune responses and metastasis in triple negative breast cancer [69]. Research on 4T1 and E0771 mouse BC cell lines with stable PIP expression showed minimal impact on in vitro behaviors. In vivo experiments using the 4T1 syngeneic mouse model revealed PIP-expressing tumors exhibited delayed onset and reduced growth, tied to increased natural killer cells and decreased type 2 T-helper cells. Interestingly, PIP expression was paradoxically associated with elevated lung metastatic colonies, possibly due to upregulated metastasis-related genes in PIP-expressing cells. This suggests PIP's dual role in BC, affecting both antitumor immunity and metastasis.

4)     RUNX2 is a myeloid-macrophage specific transcription factor. It is not expressed in mammary or other epithelial cells. The RUNX2 dependent mechanism of PIP regulation described on pages 5-6 is not relevant to its role in breast carcinoma epithelial cells. In context of tumors, it is only relevant to osteosarcoma (line 197) because osteoclasts are bone macrophages that do express RUNX2. The presented information implies that RUNX2 regulates PIP expression in BC cells which is not very likely. Unless authors can cite a reference that shows protein expression of RUNX2 (not mRNA) in human breast cancer, the described mechanism should be cited with some caution. 

Thank you for your comment.  We have revised the manuscript and added more information: 

In the article by Inman et al., it was shown that the Runx2 protein is expressed in several mammary epithelial cell lines and in primary mammary epithelial cells. Runx2 protein was detected by Western blot analysis of nuclear extracts. The study found that Runx2 expression is not restricted to metastatic human breast cancer cell lines (MDA-MB-231) but is also clearly expressed in non-metastatic human breast cancer cells (MCF-7), non-tumorigenic (HC11) cells, and primary mammary epithelial cells [54].

In the article by Selvamurugan et al., the Runx2 protein was also detected by Western Blot in mammary epithelial cells (MDA-MB-231) [55].

In the study by Onodera et al., one hundred and thirty-seven formalin-fixed and paraffin-embedded breast cancer specimens were used in this analysis of immunohistochemical study. Immunoreactivity was evaluated using the labeling index (LI). Runx2 immunoreactivity was detected in both carcinoma and stromal cells, as well as non-pathological ductal cells. The nuclear LI of Runx2 in carcinoma cells was associated with the clinical stage, histological grade and HER2 status of the patients examined. In addition, among the patients not associated with distant metastasis, those with high Runx2 LI demonstrated a significantly worse clinical outcome than those with a low LI [56].

The previously established knowledge indicated that androgens induce PIP expression in different breast cancer cell lines such as T47D, ZR-75, and MDA-MB-453 [7,57]. In a study led by Baniwa et al., a novel mechanism by which androgens stimulate PIP through Runx2 was elucidated for the first time [58]. The investigation encompassed human breast cancer T47D cells and prostate cancer C4-2B cells, highlighting the pivotal roles of AR, Runx2, and dihydrotestosterone (DHT) in this pathway. 

Notably, both AR and Runx2 transcription factors exhibit specific sequences preceding the PIP transcription start site.In analysis done by Sandelin et al. four distinct regions were identified and designated as R-I (-0.9-kb), R-II (-2.4-kb), R-III (-9.4-kb), and R-IV (-11-kb) [59]. Regions I and II serve as binding sites for Runx2, while regions III and IV act as binding sites for both Runx2 and AR. “

5)     Reference 59 is cited as the published evidence of PIP expression in 90% of BC but this paper from 1999 does not contain any images of stained tumors, only PCR data. The authors need to determine the quality and validity of cited evidence before distributing it for further dissemination in a review article. Other studies determined much lower percentage of PIP positive tumor (e.g., 39% in reference 66), and Human Protein Atlas shows extremely limited expression in 1-2 well-differentiated tumors with zero expression in multiple de-differentiated cancers. It is entirely possible that loss of differentiation is a cause for poor prognosis, not downregulation or reduced expression of PIP. The statement on lines 277-278 might be similarly interpreted – tumors that still retain tissue-specific architecture might have better prognosis, which may or not have any relevance to expression of PIP.

Thank you for your comment, we have expanded this answer and added newer sources that indicate lower expression. We have shown the differences shown in several different papers:

Some studies indicate that he prolactin inducible protein (PIP) is expressed to varying degrees in more than 90% of breast cancers (BCs) [68,69]. However, in more recent publications, the information regarding expression indicates slightly lower levels of PIP protein. In study conducted by Darb-Esfahani et al. only 39.7% breast tumors were PIP postitive [70]. Similarly, a recent study by Shiran et al. observed PIP expression in 20 out of 60 breast cancer patients (33.3%)  [71]. In another paper histopathologic testing on apocrine carcinoma of the breast revelaed that 60% of tumour were PIP positive  [72].

6)     Conclusions state that PIP is a highly promising therapeutic factor but no published evidence is cited to support this statement. The cited papers noted only correlation (not causation) between lower PIP and higher grades, which could be explained by coincident downregulation of PIP in de-differentiating tumor cells. That does not mean that PIP inhibits progression of malignancy, and this certainly does not mean that upregulation of PIP can prevent or suppress development of malignancy (no evidence for a therapeutic effect). It is also unlikely to play an anti-tumor role and at the same time be a marker of metastasis. There is also very little evidence that PIP can be used for anti-tumor vaccination strategy, particularly in the light of its low expression in tumors other than early-staged and highly differentiated BC. There is no doubt that PIP is an interesting protein but conclusions should be a little bit more objective with regard to its actual clinical significance in BCa.

Thank you for your comment, we have changed the title and conclusions and focused only on the correlation of pip protein expression with tumorigenesis in breast cancer. Indeed, it may be more valuable to highlight only the prognostic role.

Minor critiques:

1)     Line 48 – define ZAG

Thank you for your comment, we have defined ZAG in the manuscript: 

ZAG (.Zinc 2-glycoprotein)”

2)     Line 80 – define or edit “suggestive significance”

Thank you for your comment, we have edited this in the  manuscript:

There are also some evidences in gene-based analyses about role of PIP genesignificance in association with anxious temperament and unipolar depression [20]. 

3)     Line 199 – should be MDA-MB-453

Thank you for your comment, we have corrected this error. 

4)     Line 203 – the sentence needs to be edited for proper grammar

Thank you for your comment, we have edited this sentence. 

5)     Line 264 – should be ER-negative

Thank you for your comment, we have edited this in the manuscript.

6)     Line 270 – should be cytokeratins 5, 6

Thank you for your comment, we have edited this in the manuscript.

7)     Line 366 – should be breast cancer

8)     Last name of authors can be included but no need for the first name (line 200) and/or first initial (line 224)

Thank you for your comment, we have edited this in the manuscript.

9)     Define “molecular apocrine” breast cancer

Molecular apocrine (MA) tumors are estrogen receptor (ER) negative breast cancers characterized by androgen receptor (AR) expression. This is defined in the manuscript: “Another regulating mechanism of the PIP gene was found in a subtype of ER-negative, AR-positive breast cancer called molecular apocrine.”

In conclusion, we would like to express our gratitude for your diligent review, which has significantly contributed to the refinement of our manuscript. Your insights have been instrumental in enhancing the accuracy and clarity of our work. We are hopeful that the revised manuscript aligns more closely with the journal's standards and addresses the concerns raised. We look forward to the possibility of having our work accepted for publication.

Reviewer 2 Report

The manuscript by Sauer et al., is focusing on the role of Prolactin-inducible protein (PIP) in breast cancer. The authors emphasized the structure of PIP and the mechanisms that regulate its expression. Also, they summarize the research around of the prognostic significance of PIP expression in breast cancer, previously to describe its impact in cell-based therapies. The manuscript is well written, and it is described with the logical flow.

Here my commentaries:

1.       In my opinion, a brief introduction should be included with a clear description of the aim and importance of this report beyond what other papers on the same topic have previously identified. What is the original contribution?  

2.       The 35% of references were published before 2000 year. I encourage the authors use recent references to support the discussion of their manuscript.

3.       I suggest to include an image of the 3D structure of PIP that allow the understanding of the description in point 1.

4.       Could authors edit the information included in point 3 and 5? This information can be integrated in one subhead.

5.       Reference section should be edited according with guides of this journal.

Author Response

Dear Reviewer,

We sincerely appreciate your thorough review and insightful comments on our manuscript. Your feedback has been immensely valuable in refining our work. We have carefully considered each of your points and made the necessary revisions to address your concerns. 

Below are the original comments and our corresponding responses:

  1.       In my opinion, a brief introduction should be included with a clear description of the aim and importance of this report beyond what other papers on the same topic have previously identified. What is the original contribution?

Thank you for your comment. We have added a brief introduction which highlights aim of the paper: 

“  1. Introduction

In 1986, Haagensen et al. provided the initial description of this acidic protein, which was notably abundant in the gross cystic disease fluid of the breast [1,2]. The Prolactin-inducible protein (PIP), also known as gross cystic disease fluid protein 15 (GCDFP-15), has emerged as a subject of significant interest in recent years due to its potential role as a specific marker in breast cancer.This protein, encoded by the PIP gene, has been extensively studied for its diverse functions and intriguing characteristics. From its unique structural features to its involvement in regulatory pathways and clinical implications, PIP has captured the attention of researchers in the field. In the context of breast cancer, the expression of PIP holds clinical significance [3].  It varies depending on the tumor stage and subtype [4].  PIP's role in response to chemotherapy highlights its potential as a predictive biomarker for treatment outcomes and currently being investigated in clinical trials [5].  In this paper, we delve into the structural features, regulatory mechanisms, clinical implications, and potential therapeutic applications of PIP in breast cancer. We provide insights into its diverse functions, its impact on tumor progression, and its potential as a candidate for innovative cell-based therapies. Through a comprehensive exploration of the multifaceted roles of PIP, we aim to contribute to a deeper understanding of its significance in breast cancer research and treatment.”

  1.   The 35% of references were published before 2000 year. I encourage the authors use recent references to support the discussion of their manuscript.

Thank you for your comment. I agree that a substantial number of references are from before the year 2000. However, it's worth considering that the PIP protein was initially characterized in the 1980s and didn't attract significant interest until later. It's only in recent years that clinical studies have emerged, investigating its prognostic role in breast cancer and its response to chemotherapy. That's why we aimed to cite the pioneering works of the authors who first described its functions, rather than duplicating information from more recent articles. In conducting the review, we did include several dozen new studies as well.

  1.       I suggest to include an image of the 3D structure of PIP that allow the understanding of the description in point 1.

Thank you for your comment. We have incorporated an image depicting the 3D structure of PIP to aid in the understanding of the description in point 1

  1.       Could authors edit the information included in point 3 and 5? This information can be integrated in one subhead.

Thank you for your comment. We appreciate your feedback and have made the suggested changes. The information from points 3 and 5 has been integrated under a single subheading for better organization and clarity.

  1.       Reference section should be edited according with guides of this journal.

Thank you for your feedback. We greatly appreciate your comment. We have revised the reference section to align it with the guidelines of this journal once again. 

In conclusion, we would like to express our gratitude for your diligent review, which has significantly contributed to the refinement of our manuscript. Your insights have been instrumental in enhancing the accuracy and clarity of our work. We are hopeful that the revised manuscript aligns more closely with the journal's standards and addresses the concerns raised. We look forward to the possibility of having our work accepted for publication.

Round 2

Reviewer 1 Report

The authors addressed all raised issues in my previous critique. 

Reviewer 2 Report

This manuscript has been appropriately edited and refined by authors. My main concerns have been carefully addressed.